# Oviduct Transcriptomic Reveals the Regulation of mRNAs and lncRNAs Related to Goat Prolificacy in the Luteal Phase

**DOI:** 10.3390/ani12202823

**Published:** 2022-10-18

**Authors:** Zhipeng Sun, Qionghua Hong, Yufang Liu, Chunhuan Ren, Xiaoyun He, Yanting Jiang, Yina Ouyang, Mingxing Chu, Zijun Zhang

**Affiliations:** 1College of Animal Science and Technology, Anhui Agricultural University, Hefei 230036, China; 2Key Laboratory of Animal Genetics, Breeding and Reproduction of Ministry of Agriculture and Rural Affairs, Institute of Animal Science, Chinese Academy of Agricultural Sciences, Beijing 100193, China; 3Yunnan Animal Science and Veterinary Institute, Kunming 650224, China

**Keywords:** goat, oviduct, luteal phase, fecundity, mRNA, lncRNA

## Abstract

**Simple Summary:**

The kidding number is an important reproductive trait in domestic goats. The oviduct, as one of the most major organs, is directly involved in the reproductive process, providing nutrition and a location for early embryonic development. The current study provides genome-wide expression profiles of mRNA and long noncoding RNAs (lncRNAs) expression in Yunshang black goat, a new breed of meat goat bred in China with a high kidding number. During the luteal phases, oviduct mRNAs and lncRNAs associated with high- and low-fecundity Yunshang black goats were identified, and their potential biological functions were predicted using GO, KEGG, and GSEA enrichment analysis. These findings shed light on the oviduct-based prolificacy mechanism in goats.

**Abstract:**

The oviduct is associated with embryo development and transportation and regulates the pregnancy success of mammals. Previous studies have indicated a molecular mechanism of lncRNAs in gene regulation and reproduction. However, little is known about the function of lncRNAs in the oviduct in modulating goat kidding numbers. Therefore, we combined RNA sequencing (RNA-seq) to map the expression profiles of the oviduct at the luteal phase from high- and low-fecundity goats. The results showed that 2023 differentially expressed mRNAs (DEGs) and 377 differentially expressed lncRNAs (DELs) transcripts were screened, and 2109 regulated lncRNA-mRNA pairs were identified. Subsequently, the genes related to reproduction (*IGF1*, *FGFRL1*, and *CREB1*) and those associated with embryonic development and maturation (*DHX34*, *LHX6*) were identified. KEGG analysis of the DEGs revealed that the GnRH- and prolactin-signaling pathways, progesterone-mediated oocyte maturation, and oocyte meiosis were related to reproduction. GSEA and KEGG analyses of the target genes of DELs demonstrated that several biological processes and pathways might interact with oviduct functions and the prolificacy of goats. Furthermore, the co-expression network analysis showed that XLOC_029185, XLOC_040647, and XLOC_090025 were the cis-regulatory elements of the DEGs *MUC1*, *PPP1R9A*, and *ALDOB*, respectively; these factors might be associated with the success of pregnancy and glucolipid metabolism. In addition, the *GATA4*, *LAMA2*, *SLC39A5*, and *S100G* were trans-regulated by lncRNAs, predominantly mediating oviductal transport to the embryo and energy metabolism. Our findings could pave the way for a better understanding of the roles of mRNAs and lncRNAs in fecundity-related oviduct function in goats.

## 1. Introduction

The high fecundity of goats is a key economic feature that has a direct impact on the production efficiency of the goat farming industry. Many goat breeds have two kids, limiting the availability of goats for meat, fur, and milk products [1]. The oviduct is an important conduit between the ovary and the uterus, and its internal structure changes on a regular basis due to estrogen and progesterone [2]. These modifications primarily involve changes in the ratio of multiciliated cells (MCs) to secretory cells [3]. These modifications create the ideal microenvironment for gamete maturation, sperm capacitation, fertilization, and early embryo development [4,5]. The proportion of oviduct secretory cells increases significantly during the luteal phase, which is crucial for the preimplantation development of the embryo [6]. This highlights the significance of the oviduct in goat reproduction. Numerous previous studies have revealed a large number of reproduction-related genes in the ovary, uterus, and hypothalamic–pituitary–gonadal (HPG) axis involved in the regulation of folliculogenesis and oocyte maturation [7,8]. However, oviduct research is largely unresolved. As a result, elucidating the molecular regulatory mechanism of goat reproduction in the oviduct in the luteal phase will provide a novel perspective for the successful pregnancy and molecular breeding of she-goats.

In the past few decades, the breeding of new varieties of domestic animals has undergone a cross-field transition from phenotypic selection to molecular breeding. It seems that the selection of animals with a high ovulation rate and litter size characteristics based on their genotype results in higher breeding efficiency. However, many factors restrict litter sizes, which are mainly affected by heredity and some crucial gene mutations in addition to external factors [9,10]. For example, the SNPs of acetylserotonin O-methyltransferase (*ASMT*) and ADAM metallopeptidase with thrombospondin type 1 motif 1 (*ADAMTS1*) are significant for the litter size, and the single-strand conformation polymorphism (*SSCP*) of bone morphogenetic protein 15 (*BMP15*) may be related to the prolificacy of Jining Grey goats [11,12]. In addition, several transcription factors, genes, some signaling pathways, and even noncoding RNAs (ncRNAs) are commonly involved in reproductive regulation [13]. For example, *BMP15*, growth differentiation factor 9 (*GDF9*), and bone morphogenetic protein receptor 1B (*BMPR1B*) are all prolificacy-related genes in goats [14]. In addition to these major genes in the whole-transcriptome sequencing, noncoding RNAs (ncRNAs) are also widely studied [15,16,17]. 

Long noncoding RNAs (lncRNAs) are a type of ncRNA with a length longer than 200 nucleotides [18]. These RNAs regulate gene expression by modulating gene transcription, precursor mRNA splicing, and transport [19,20], and most studies suggest that lncRNAs act as *cis* or *trans* factors in the epigenetic and transcriptional regulation of nearby genes [21,22]. In mammals, studies on the regulation of reproduction by lncRNAs have been reported, such as lncRNA_PVT1 regulating the E2 and E4 secretion, proliferation, and apoptosis of PCOS ovarian granulosa cells [23]. LncGDAR was demonstrated to facilitate ovarian granulosa cell (GC) apoptosis by regulating the apoptosis-related genes in sheep [24]. LncRNA TUNAR may involve in embryo implantation by modulating the blastocyst attachment to the endometrial epithelium and regulating the proliferation and decidualization of ESCs [25]. Meanwhile, germ cell formation, early embryo implantation and development, and hormone regulation in human and animal reproduction are regulated by lncRNAs [18,26]. In recent years, lncRNAs have been shown to play key roles in the regulation of ovulation, oocyte maturation, and embryo transfer in mammals’ reproductive processes [17,25,27]. In addition, genome-wide analysis studies were performed in goat ovary and uterus and showed that XR_001917388.1 and TUCP_000849 may play a role in oogenesis and oocyte maturation in goats [28], and lncRNAs LNC_007223, LNC_005256, and LNC_010092 may contribute to the implantation of the embryo [29]. Although there have been several studies on lncRNAs as reproduction-related regulatory factors [16], little research on the transcriptome profile of the complete oviduct in fertile she-goats under specific physiological conditions. 

The key stage of gamete and embryo development mainly occurs primarily in oviduct in the luteal phase [30], which has a direct impact on the kidding numbers and multiple births. Using the biological relationship between lncRNA and mRNA, we attempted to investigate the oviductal factors influencing goat fecundity. Specifically, we employed high-throughput RNA sequencing (RNA-seq) to look for changes in mRNA and lncRNA expression profiles in the oviduct. Briefly, a total of 2023 DEG and 377 DEL transcripts were identified between the two groups. We projected likely *cis* or *trans* targeting links of lncRNA-mRNA pairs. After RT-qPCR validations for four DEGs and five DELs, the RNA-seq was shown to be reliable. Then, Gene Ontology (GO), Kyoto Encyclopedia of Genes and Genomes (KEGG), and Gene Set Enrichment Analysis (GSEA) were used to identify the functions of the differentially expressed mRNAs and the target genes of the differentially expressed lncRNAs. We then constructed regulatory networks to further understand the key role of lncRNAs connected to prolificacy. Our findings laid the groundwork for prolific goat lncRNA and mRNA potential relationships in the regulation of goat kidding numbers.

## 2. Materials and Methods

### 2.1. Ethics Statement

All the related experiments involving goats and the protocols were approved by the Science Research Department (in charge of animal welfare issues) of IAS-CAAS (Beijing, China). Ethical approval was given by the Animal Ethics Committee of the IAS-CAAS (No. IAS 2021-23). All goats were from the original breeding farm of Yunshang black goat (Yunnan Province, China), where they all had the same feeding conditions.

### 2.2. Experimental Animals and Sampling

Tissue samples of the oviducts from ten nonpregnant adult female Yunshang black goats (three years old) showed no significant differences in weight, height, and age. Animals were obtained from Yixingheng Animal Husbandry Technology Co., Ltd. Tuanjie Township Base in Kunming City (Yunnan, China). All goats were raised under the same conditions and divided into two groups: the high-fecundity group (*n* = 5, average kidding number 3.4 ± 0.42, LH group) and the low-fecundity group (*n* = 5, average kidding number 1.8 ± 0.27, LL group). Before the experiment, all nonpregnant goats were subjected to simultaneous estrus based on progesterone vaginal suppository (CIDR), which was removed after Day 16; the removal time was set as 0 h. The 10 goats were slaughtered within 168 h after CIDR removal (luteal phase; the average corpus luteum number of LH and LL groups was 3.6 ± 0.40 and 1.6 ± 0.24, respectively). Oviduct samples were collected and immediately snap-frozen in liquid nitrogen and stored at −80 °C until RNA extraction.

### 2.3. Total RNA Extraction and Detection

The total RNA of each oviduct sample was extracted using TRIzol (Invitrogen, Carlsbad, CA, USA) and DNase I (Qiagen, Beijing, China) according to the manufacturer’s protocols. The purity and concentration of the isolated RNA were quantified using a NanoDrop 2000 spectrophotometer (Thermo Scientific, Wilmington, DE, USA), and the integrity was detected by 1.5% agarose gel electrophoresis. An Agilent 2100 Bioanalyzer (Agilent Technologies, Palo Alto, CA, USA) with an RNA Nano 6000 Assay Kit was used to assess the integrity of total RNA. All samples had an RNA integrity number (RIN) ranging from 8.0 to 9.2, and the value of samples greater than 8.0 was considered acceptable for RNA-seq.

### 2.4. Library Preparation, RNA-seq, and Data Quality Control 

A total of ten cDNA libraries of the oviducts in the luteal phase were constructed. Briefly, 3 micrograms of RNA per sample was used as input material for rRNA (ribosomal RNA) removal, using the Epicentre Ribo-Zero^TM^ rRNA Removal Kit (Epicentre, Madison, WI, USA). The libraries were constructed using the NEB Next^®^ Ultra™ Directional RNA Library Prep Kit for Illumina^®^ (NEB, Beijing, China), according to the manufacturer’s instructions, and index codes were used to label the sequences of each sample. RNA-sequencing libraries were generated by paired-end sequencing. Finally, the Agilent Bioanalyzer 2100 system was used to assess library quality, and the AMPure XP system was used to purify the products. Subsequently, the pooled libraries were sequenced on the Illumina Novaseq platform using a chain-specific library construction strategy to count the numbers and types of various transcripts (mRNAs, known lncRNAs, and novel lncRNAs). 

To further ensure the quality of the analytical results, total paired-end sequence reads were checked using in-house scripts to obtain high-quality reads. After removing the reads containing >10% poly-N bases accounted for >1% and low-quality reads (≤20) from the raw data, the Illumina sequencing raw reads were obtained, among which the number of bases with a quality value Q ≤ 20 was >50%. Simultaneously, the Q20, Q30, and GC contents of the clean data were calculated, and all high-quality downstream areas were analyzed. Immediately, HISAT2 (v.2.1.0) [31] was used to map the clean reads of each sample to the reference genome (GCF_001704415.1). Only the uniquely mapped reads were assembled, and the expression levels were predicted using String Tie software (v.1.3.5) [32]. Next, the FPKM (fragments per kilobase of exon model per million mapped fragments) [33] for each gene was obtained. Finally, we calculated the number and ratio of uniquely mapped reads within the three gene functional elements (exons, introns, and intergenic elements). Thus, known mRNA and lncRNA transcripts were identified, and the positions of the transcripts were determined.

### 2.5. LncRNA Identification and Differential Expression Analysis

To reduce the false-positive rate, candidate lncRNAs (intronic lncRNA, long intergenic noncoding RNA (lincRNA), and antisense lncRNA) were screened based on the locations of coding reads, and the next step was to calculate the coding potential. Novel lncRNA candidates were identified based on a length ≥200 bp and >1 exon number. Meanwhile, the CUFFcompare program was used to filter transcripts with overlapping regions of mRNA and other noncoding RNAs (rRNA, tRNA, etc.) from a known database, and transcripts annotated as “I,” “j,” “o,” “u,” and “x,” representing lncRNAs of potentially novel intronic, potentially novel isoform with more than one splice junction of a reference transcript, generic exonic overlap with a reference transcript, potentially intergenic, and potentially antisense transcripts, respectively, were kept. Next, CNCI [34], CPC2 (http://cpc2.cbi.pku.edu.cn/, accessed on 6 January 2022) [35], and PLEK [36] coding potential prediction software programs were used to evaluate the protein-coding potential of the transcripts. The mRNA and lncRNA expression levels were estimated using the fragments per kilobase per million mapped reads (FPKM) values. DESeq2 [37] was used to calculate log_2_(fold change) and *p* value based on the normalized counts. The thresholds for screening significantly differentially expressed (DE) mRNAs and lncRNAs were an adjusted *p* value < 0.05 and |log_2_(fold change)| ≥ 1. Subsequently, the [log_2_ (FPKM)] values of each mRNA and lncRNA were systematically clustered using pheatmap (v.1.0.2) to analyze the resemblance and relationships between different libraries.

### 2.6. Quantitative Real-Time PCR

Ten DEGs and DELs were selected to validate the accuracy of RNA sequencing via the real-time quantitative polymerase chain reaction (RT-qPCR), respectively. The primers were designed with Primer Premier 6 and synthesized by Sangon Biotech (Beijing, China) (Appendix A), and *RPL19* was used as the reference gene. For RT-qPCR analysis, cDNA (complementary DNA) was obtained from the reverse transcription of 1000 ng RNA using the PrimeScript^TM^ RT regent kit (TaKaRa, Beijing, China). RT-qPCR was performed using the TB Green^®^ Premix Ex Taq™ II (TaKaRa, Beijing, China) according to the manufacturer’s instructions, and all genes and lncRNAs were analyzed in triplicate. RT-qPCR was then performed on a QuantStudio^®^ 3 (ABI, Foster City, CA, USA). The 2^−ΔΔCt^ method was used to calculate the relative gene and lncRNA expression levels. All the RT-qPCR results are presented as the mean ± SEM, and a *p* value < 0.05 was considered statistically significant.

### 2.7. Target Gene Prediction of DELs and Functional Bioinformatics Analyses

The potential protein-coding genes were predicted to be *cis*- and *trans*-acting based on the lncRNA locus, the 100 kb downstream and upstream protein-coding genes (without overlap) were first identified as *cis*-acting target genes, and the genes that overlapped with the predicted lncRNAs predicted were selected as the *trans*-acting target genes. Pearson’s correlation coefficient was used to determine the association of lncRNAs and mRNAs through their expression, and we selected DELs and DEGs to establish a coexpression network using Cytoscape software (v.3.9.0, the Cytoscape Consortium, San Diego, CA, USA).

The DEGs and the targeted genes of lncRNAs were statistically enriched and classified by GO annotation and KEGG pathway analyses were performed using the online software Metascape (https://metascape.org/gp/index.html#/main/step1, accessed on 20 July 2022) with default parameters to explore the potential functions of all DGEs and DELs target genes. GO terms were classified into cellular components (CC), molecular functions (MF), and biological processes (BP), and only the enriched terms were considered statistically significant with a corrected *q* value < 0.05. The KEGG pathways with a significance threshold of *p* value < 0.05 were considered. To further illustrate the role of protein-coding genes in female goat reproduction, a PPI network was constructed using the STRING database (https://string-db.org/; Organism: *Ovis aries;* accessed on 6 January 2022). GSEA was implemented by the GSEA software (v.4.2.1) with default parameters to analyze the predicted target genes of lncRNAs from the biological process ontology of C5 (ontology gene sets). We also visualized the interaction network, depending on the *cis*- or *trans*-acting interactions, using Cytoscape software (v.3.9.0).

### 2.8. Statistical Analysis

LncRNAs and mRNAs with *p* values of < 0.05, which were considered to be differentially expressed. The results of the RT-qPCR were calculated using the 2^−ΔΔCt^ method [38]. All data are presented as the means ± SEMs of three replicates, and statistical significance was represented by a *t*-test (*p* value < 0.05). The GraphPad Prism (version 9.3.1) software (San Diego, CA, USA) was used for statistical analyses.

## 3. Results

### 3.1. Summary of Sequencing Data in YunShang Black Goat Oviducts in the Luteal Phase

Ten oviduct samples from the two groups were used to construct ten cDNA libraries for sequencing. The RNA-seq data were subjected to quality control. After removing the low-quality, adapter-containing, and containing poly-N sequence reads from the raw reads, more than 50 million clean reads were obtained. The average data for Q20 was 97.85% and 98.34%, the Q30 was 93.90% and 94.90%, and the average GC content was 45.33% and 46.54% in the LH and LL groups, respectively. To verify the reliability of the sequencing results, HISAT2 (29) was used to compare and analyze the reference genome of the clean reads. The average ratios of total mapped reads were 96.69% and 97.26% and uniquely mapped reads were 93.18% and 93.25% in the LH and LL groups, respectively (Table 1).

### 3.2. Identification of mRNA and lncRNA in YunShang Black Goat Oviduct Tissue

After mapping the reference, 24,170 lncRNAs were identified in the oviducts of ten goats using CNCI, CPC2, and PLEK tools, as shown in Figure 1A (Appendix A); approximately 3297 were antisense-lncRNAs (13.6%), 8187 were lincRNAs (36.5%) and 12,056 were intronic-lncRNAs (49.9%) (Figure 1B, Appendix A). In addition, 43,779 mRNAs and 24,370 novel mRNA transcripts were identified (Appendix A). The length distribution of the lncRNAs was approximately 200–600 bp, mRNAs were mainly distributed from 800 to 4000 bp, and more than 38.63% of mRNAs had a length distribution exceeding 4000 bp (Figure 1C). These lncRNAs and mRNAs were randomly assigned to 29 autosomes and X-chromosomes, and their distributions were similar (Figure 1D). Interestingly, one lncRNA and thirteen mRNAs were found on the mitochondria, suggesting that the lncRNAs and mRNAs may be involved in cytoplasmic biological functions. We also found that approximately 1070 lncRNAs (4.12%) and 2241 mRNAs (3.29%) were not localized to any chromosome, and 1 lncRNA and 13 mRNAs were found to align to a mitochondrial location. Meanwhile, most lncRNA transcripts had two to three exons, and mRNA transcripts covered a wide range from two to 40, while 4% of the transcripts exceeded 65 exons, which is significantly more than the average of lncRNA transcripts (Figure 1F). Regarding the mRNA expression level, only 0.82% were highly expressed with FPKM > 60, and the expression levels of most genes were at 0~1 FPKM in all ten tissue samples (Appendix A). Meanwhile, the expression profiles of mRNAs and lncRNAs were further analyzed based on Log_10_ (FPKM+1). The expression levels of most lncRNA and mRNA transcripts were <1, and the number of lncRNA and mRNA transcripts decreased with increasing FPKM. The results showed that the number of lncRNA transcripts was higher than that of most mRNA transcripts (Figure 1E).

### 3.3. Profiling of Differentially Expressed Transcripts and qPCR Verification

In total, 68,149 mRNA transcripts and 25,958 lncRNA transcripts were selected in the oviduct. Based on the criteria of |log_2_ FC (fold change)| > 1 and *q* value *<* 0.05, 2024 DEG transcripts (824 genes) were identified between the LL and LH groups, of which 1077 were upregulated and 947 were downregulated, including 47 new genes (Figure 2A, Appendix A). Furthermore, 377 DELs were screened, of which 210 were upregulated and 167 were downregulated between the two groups, and two known lncRNAs and two new lncRNAs were identified, as shown in Figure 2B (Appendix A). The cluster heatmap of the DEGs and DELs revealed the expression patterns in the LL and LH oviducts of goats (Figure 2C,D, Appendix A). The top 50 significantly different genes in the low- and high-fecundity groups are shown in Figure 2E. The five most highly expressed genes in the low fecundity groups were *ZNF75D*, *HS6ST2*, *SEMA3F*, *SCAMP1*, and *PTGER2*, and six novel DEGs (*LOC102177580*, *LOC102174380*, *LOC102180173*, *LOC108635614*, *LOC108638473*, and *LOC102186806*) were included; the five most highly expressed genes in the high fecundity groups were *ATP9B*, *LOC102173506*, *KLHL31*, *KCTD20*, and *TNIP1*, and another eight new DEGs (*LOC102170691*, *LOC108635304*, *LOC108636024*, *LOC102175889*, *LOC102183132*, *LOC102169846*, *LOC102168342*, *LOC102175560*, and *LOC08634436*) were identified. To validate the accuracy of the RNA-seq, ten DEGs (*OVN*, *SPOCK1*, *KRT7*, *AZGP1*, *KREMEN1*, *APOD*, *DPT*, *POSTN*, *ZNF75D* and *PID1*) and ten DELs (XLOC_104485, XLOC_050288, XLOC_239472, XLOC_213822, XLOC_173492, XLOC_200842, XLOC_171602, XLOC_017318, XLOC_055290 and XLOC_023225) were randomly selected for RT-qPCR validation with three biological replicates and compared with the RNA-seq data. The results showed consistent expression levels (Figure 3A,B), suggesting that the RNA-seq data are highly accurate.

### 3.4. PPI Network Related to Prolificacy Traits

To investigate the functions of differentially expressed protein-coding genes, we employed the online search tool STRING (https://string-db.org, accessed on 20 July 2022) and based on a score of 0.9 to shed light on a protein-protein interaction (PPI) network for the functional characterization of the annotated proteins. A total of 218 nodes and 202 edges comprised the network. Based on gene function, we identified a core network, and *PTK2*, *MAPK8IP3*, *PTPN13*, *CREB1*, *CAMK2D*, *PPIP5K1*, *PPIP5K2*, *TPM2*, and *TPM3* were the key DEGs in the networks with other proteins (Appendix A).

### 3.5. Functional Enrichment Analysis of the DEGs 

We further used GO analysis to perceive the potential of DEGs. A total of 488 GO terms (Appendix A) were significantly enriched in the LL vs. LH groups. Cell junction organization, actin filament-based process, small GTPase-mediated signal transduction, regulation of GTPase activity, and intracellular receptor were the top five terms in biological process (BP). For cellular component (CC), axon, microtubule, polymeric cytoskeletal fiber, centrosome, perinuclear region of cytoplasm, and transcription regulator complex were the most highly annotated terms. Regarding molecular function (MF), the most significant terms included transcription coregulator activity, GTPase regulator activity, nucleoside-triphosphatase regulator activity, tubulin binding, protein kinase activity, actin binding, transcription factor binding, and cell adhesion molecule binding. The top 20 GO functional annotations are shown in Figure 4A (Appendix A).

KEGG enrichment analysis revealed that 86 pathways were annotated. Most differentially expressed mRNAs were included in several significant pathways, such as adherens junction, insulin secretion, axon guidance, regulation of actin cytoskeleton, GnRH, WNT, calcium, and prolactin signaling pathway, ubiquitin-mediated proteolysis, mitophagy-animal, progesterone-mediated oocyte maturation, fructose and mannose metabolism, oocyte meiosis, and the results are summarized in Figure 4B. It is noteworthy that insulin secretion, circadian entrainment, GnRH signaling pathway, progesterone-mediated oocyte maturation, oocyte meiosis, and prolactin signaling pathway were directly associated with reproduction; the rest of the pathways are shown in Appendix A.

### 3.6. Posttranscriptional Regulatory Network of DELs and the Target Genes

DEL transcripts and the target genes (target DEGs) were selected to construct an mRNA-lncRNA coexpression network, as shown in Appendix A. This network consisted of 475 nodes and 2109 edges, of which one mRNA was associated with multiple lncRNAs or one lncRNA was associated with multiple mRNAs, indicating that there was mutual regulation between lncRNAs and mRNAs. For example, one known lncRNA, LOC108637846, corresponded to 25 target *trans*-acting protein-coding genes (Appendix A). The features of the top 50 target genes are shown in the heatmap for each phenotype (Figure 5B).

Furthermore, KEGG pathway analysis revealed that these DEL target genes were enriched in significant pathways, such as axon guidance; adherens junction; WNT signaling pathway; mTOR signaling pathway; AMPK signaling pathway; phosphatidylinositol signaling system; tight junction; and valine, leucine, and isoleucine biosynthesis (Appendix A). All target genes involved in these pathways are shown in the network (Figure 5A). In addition, we used the method of GSEA to further determine the association of lncRNAs with oviduct function. GSEA indicated that highly expressed DEGs (target genes) in the LL group were functionally associated with the regulation of the mitotic cell cycle, vesicle cytoskeletal trafficking, cytokinesis, carbohydrate catabolism, the regulation of anatomical structure size, and telomere organization (Figure 5C), whereas highly expressed DEGs (target genes) in the LH group were related to positive regulation of cell death, actin filament-based movement, regulation of protein serine-threonine kinase activity, regulation of blood circulation, actin-mediated cell contraction and regulation of actin filament-based movement (Figure 5D). To better reflect the association of lncRNAs in oviductal function, the WNT signaling pathway; mTOR signaling pathway; AMPK signaling pathway; phosphatidylinositol signaling system; tight junction; and valine, leucine, and isoleucine biosynthesis were mainly associated with the function of the oviduct, and affected reproduction was selected to establish the coexpression network of DEGs and related DELs in these pathways (Figure 6).

## 4. Discussion

The morphology and function of mammalian oviducts were changed during the estrous [3], which was primarily caused by the periodic changes of the cilium and secretory cells mediated by estrogen and progesterone [39]. Secretory cells dominated the epithelial of the oviduct in the luteal phase, and secretion is an important medium of the microenvironment required to promote healthy gamete and embryo development [40]. To analyze the scientific problems of fecundity, it is critical to understand the molecular mechanism and function of the oviduct during the luteal phase. In recent years, several studies have presented evidence that lncRNAs are widely involved in the regulation of neighboring genes, biological processes, and mammalian reproduction [41]. However, few studies on lncRNAs and mRNAs in the goat oviduct have been conducted. Herein, RNA-seq was used for genome-wide analyses to reveal the mRNA and lncRNA expression profiles. We attempted to build a coexpression network to investigate the relationship between DELs and DEGs, as well as the possible role of lncRNAs in goat proliferation. In this study, 68,149 mRNA and 25,958 lncRNA transcripts were identified from goat oviduct tissues. For functional analysis, 2024 DEG transcripts (including 824 genes) and 377 DELs were screened. Our study comprehensively searched for lncRNAs and mRNAs in goat oviducts. These data provide valuable insights for locating operational lncRNAs and molecular mechanisms connected to goat prolificacy. 

Dopamine has been shown in previous studies to influence GnRH secretion and thus affect mammalian reproduction [42,43]. Zhang et al. emphasized that the different levels of GnRH were among the reasons for the different number of lambs [44]. During the estrous cycle, estrogen and progesterone affect the morphological and functional responses of the oviduct, which are critical for female fertility [39,45]. In the present study, most DEGs in the LL and LH comparison groups were significantly enriched in pathways, including GnRH, thyroid hormone, prolactin signaling pathway, dopaminergic synapse, insulin secretion, melanogenesis, and progesterone-mediated oocyte maturation. As previously reported, with the completion of fertilization, the secretion of steroid hormones changes, and the concentration of progesterone gradually increases, which provides conditions for oosperm cleavage and embryo development [40]. This means that the steroid hormone-dominated oviductal environment is an important influence on gamete maturation or reproduction. Moreover, some closely related reproduction pathways were annotated, such as the PI3K-Akt, MAPK, Rap1, mTOR, WNT, and calcium signaling pathways, circadian rhythm, oocyte meiosis, and fructose and mannose metabolism. Astoundingly, *ALDOB* [46] and *PFKFB3* [47,48] are involved in the fructose and mannose metabolism pathways, which may provide energy for early embryonic development. This means that the prolific goats require more metabolites to maintain high fecundity.

In our research, the expression levels of fecundity-related genes, including *IGF1*, *PRLR*, *FGFRL1*, *STARD9*, *DIO2*, *VEGFD*, *FSTL4*, *SMAD9*, *CREB1*, *SLC16A5*, and *SYNE1*, were significantly different in the LL and LH comparison. Among these genes, *IGF1* functions as a paracrine/autocrine growth factor during the luteal phase, as well as being involved in progesterone synthesis and natural corpus luteum degeneration [49]. Recently, in female beef cattle (*Bos taurus*), *IGF1* was found to promote embryonic development and, in turn, affect reproduction [50]. *FGFRL1* is a member of the FGFR family, and mice lacking the gene died during pregnancy, it is essential for fetal development [51]. *DIO2* is a critical enzyme catalyzing the conversion of thyroid hormones (THs) during seasonal reproduction in birds and mediates photoperiodic signal regulation of local TH concentrations in the hypothalamus [52]. A previous study found that *DHX34* degraded mRNAs harboring PTCs (premature termination codons) in zebrafish embryos, while the deletion of this protein caused severe developmental defects and reduces viability [53]. *LHX6* is a transcription factor associated with embryo implantation during embryogenesis that facilitates the interaction between the endometrium and the embryo to ensure pregnancy [54]. Interestingly, the expression of *IGF1* in the low-fecundity group was higher than that in the high-fecundity group (LH), with the opposite expression observed between LL and LH. FSHβ expression [55] and GnRH synthesis [56] were both regulated by *CREB1* (*Element Binding Protein 1*). Evidence suggests that *CREB1* can regulate *IGF1* at the transcriptional level [57]. Therefore, *IGF1* and *CREB1* negatively regulate estrogen secretion in the oviduct through negative regulation and affect the reproductive process in goats. These mechanisms are essential for the proper reproductive function of the oviduct. 

We speculate that the lncRNAs were expected to play dominant roles in the oviducts. Currently, a number of lines of research point to a connection between lncRNAs and mammalian reproduction [58,59,60]. A noncoding-coding gene expression network was constructed in order to thoroughly comprehend how lncRNAs and their target genes affect the reproductive capability of goats in the oviduct, and it revealed that 178 DELs were cis- or trans-regulatedly linked to 301 DEGs. In the network, *MUC1* is *cis* regulated by XLOC_029185, and its altered expression is related to ectopic pregnancies [61]. In the luteal phase, which is connected to embryonic development, four target genes (*ALDOB*, *GATA4*, *LAMA2*, and *PPPIR9A*), as well as the DELs, were increased in high-fecundity goats. XLOC_090025 has a *cis*-effect on *ALDOB*, regulates the insulin receptor signaling pathway and plays an important role in glucolipid metabolism [62], and activates WNT signaling in a GSK-3β-dependent mechanism [63], which should provide energy for embryonic development and migration. *PPP1R9A*, a *cis*-regulatory element of XLOC_040647, plays a key role in the early development of multiple types of tissues and affects embryonic growth and development in mice and cattle [64,65]. Meanwhile, the DEGs *GATA4* and *LAMA2* were all *trans*-regulators of the lncRNAs *XLOC_010441*, *XLOC_024459*, *XLOC_038137*, *XLOC_078787*, *XLOC_173265*, *XLOC_184119*, *XLOC_213586*, and *GATA4* is one of the highly conserved zinc-finger transcription factors that has been reported to play a key role in regulating the differentiation, growth, and survival of multiple cell types [66]. *LAMA2* is a key regulator of the inhibition of muscle atrophy [67] and may control the movement of the embryo via the oviduct. In the oviduct of high fecundity goats, the expression levels of the three DELs (XLOC 253092, LOC108637846, and XLOC 116687) and the two trans-target genes (*SLC39A5* and *S100G*) were dramatically increased, and the expression trends were also consistent. The DEG *SLC39A5* was *trans*-regulated by *XLOC*_*253092* and *LOC108637846*, which is primarily involved in glucose sensing and insulin secretion and regulates energy metabolism, has been identified as a candidate gene for litter size in Yunshang black goats [68]. Furthermore, *S100G* is a *trans*-regulatory element of XLOC_116687 and a member of the *S100* family, which is widely expressed in the reproductive tracts of female mammals; it is involved in the genomic effects of estradiol and is a potential candidate gene involved in facilitating oocyte and embryo transport in mating rats [69,70]. 

According to GSEA, some target genes were mapped in the terms of regulation of mitotic cell cycle, vesicle cytoskeletal trafficking, cytokinesis, carbohydrate catabolic, regulation of anatomical structure size, and telomere organization, which were the most significant biological processes in the LL groups (Figure 6C), and the terms positive regulation of cell death, actin filament-based movement, regulation of protein serine-threonine kinase activity, regulation of blood circulation, actin-mediated cell contraction and regulation of actin filament-based movement were the most enriched in the LH groups (Figure 6C). That is, most of these biological processes regulated by lncRNAs are involved in oviductal transport and energy metabolism. Moreover, we constructed pathway-DEGs-DELs regulated networks. In the network, the WNT signaling pathway and valine, leucine, and isoleucine biosynthesis were regulated by lncRNAs and indirectly regulated reproduction-related processes. The WNT signaling pathway enriched several targeted genes (*CAMK2D*, *CSNK1A1*, *TCF7L2*, *ROCK2*). Inhibiting the WNT pathway in the oviduct culture system led to shorter distances for embryo transport, according to a prior mouse study [71]. *TCF7L2*, a trans-element of the novel XLOC_203003, a high mobility group (HMG) box-containing transcription factor and plays a key role in the WNT signaling pathway response to metabolic disturbances and glucose homeostasis, and the mediation of insulin resistance [72]. Insulin is required for the regulation of the normal reproductive cycle [73]. Studies have demonstrated that infusion of a mixture of valine, leucine, and isoleucine in the late luteal phase of ewes increases the ovulation rate [74]. Notably, the upregulated gene *BIRC6* was involved in the valine, leucine, and isoleucine biosynthesis pathway, which is *trans*-regulated by *XLOC*_*010441*, *XLOC*_*024459*, *XLOC*_*038137*, *XLOC*_*078787*, and *XLOC*_*111224* and ten others upregulated lncRNAs. *BIRC6* was found to positively regulate early embryo development and blastocyst formation in bovines and mice, which are critical for a successful pregnancy [75,76]. Taken together, the DELs and their target genes identified in our study might cooperate to regulate oviductal functions for reproduction.

## 5. Conclusions

The data of the oviductal transcriptome profiles of lncRNA and mRNA in the luteal phase are valuable for revealing the prolificacy in Yunshang black goats with different fecundity. Some DEGs were involved in the regulation of estrogen secretion, embryo development, and maturation and were directly involved in the control of kidding numbers. Meanwhile, we constructed a coexpression network associated with prolificacy traits, and several lncRNAs and target genes may play key roles in the regulation of early embryo development and the function of oviduct in the luteal phase, which implied that these lncRNA–mRNA pairs function in the prolificacy regulation of the goat oviduct mechanism and will be further evaluated in future studies. Our study provides multiple potential lncRNAs and mRNAs as well as lncRNA-mRNAs for subsequent molecular assays.

## Figures and Tables

**Figure 1 animals-12-02823-f001:**
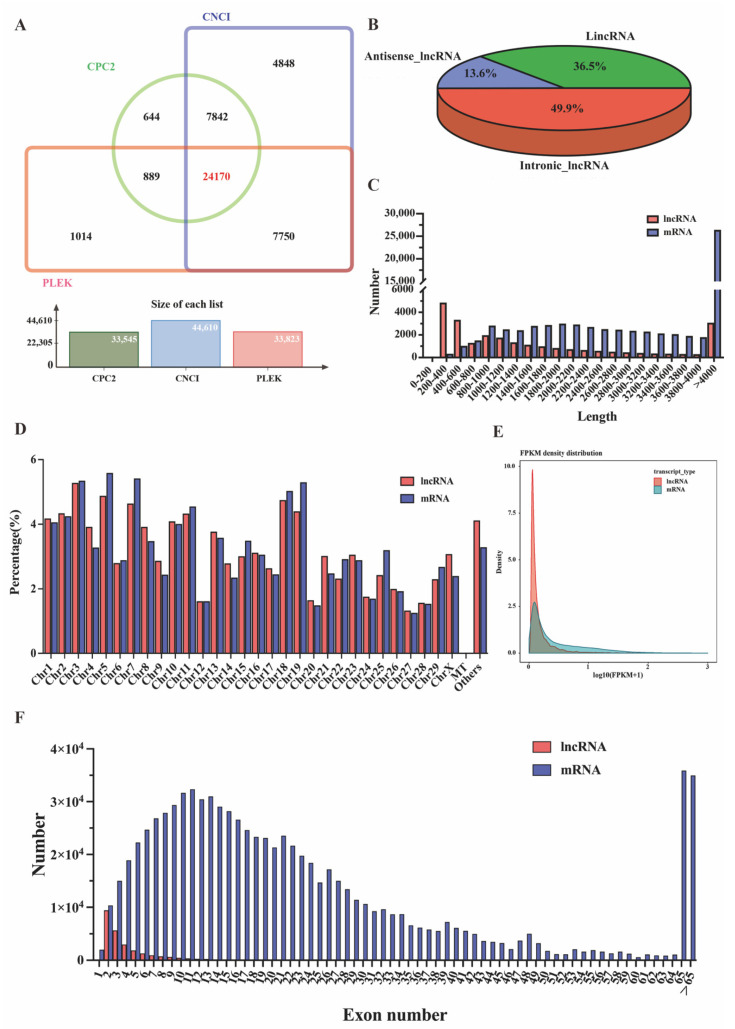
Expression profiles of lncRNA and mRNA in the oviduct. (**A**) Venn shows the common and unique number of novel lncRNAs by three methods of CNCI, CPC2, and PLEK. (**B**) Classification of novel lncRNAs, including lincRNAs, intronic-lncRNAs, and antisense-lncRNAs. (**C**) The length statistics of lncRNA and mRNA. (**D**) The distribution of lncRNAs and mRNAs in different chromosomes. (**E**) Comparison of lncRNA and mRNA expression profiles based on FPKM values. (**F**) The statistics of lncRNA and mRNA exon number.

**Figure 2 animals-12-02823-f002:**
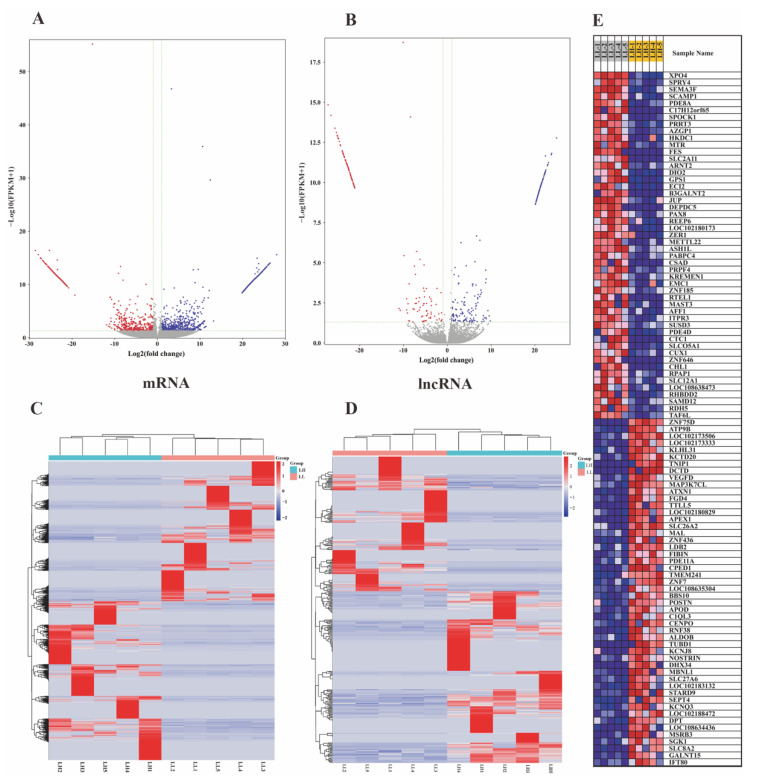
The analysis of differentially expressed mRNAs (DEGs) and lncRNAs (DELs). (**A**) Volcano plot of DEGs in LL vs. LH. (**B**) Volcano plot of DELs in LL vs. LH, of which vertical lines correspond to |log_2_ FC (fold change)| > 1 and in upregulation or downregulation; the horizontal line represents *q* value < 0.05; blue points refer to upregulated and red points refer to downregulated. (**C**) Hierarchical cluster analysis of DEGs in LL vs. LH. (**D**) Hierarchical cluster analysis of DELs in LL vs. LH. The color scale indicates log_2_(FPKM) and intensity increases from red to blue, which indicates down- and up-regulation, respectively. (**E**) The top 50 significantly differential genes in the low-and high-fecundity groups. LL stands for the low-fecundity group in the luteal phase; LH stands for the high-fecundity group in the luteal phase.

**Figure 3 animals-12-02823-f003:**
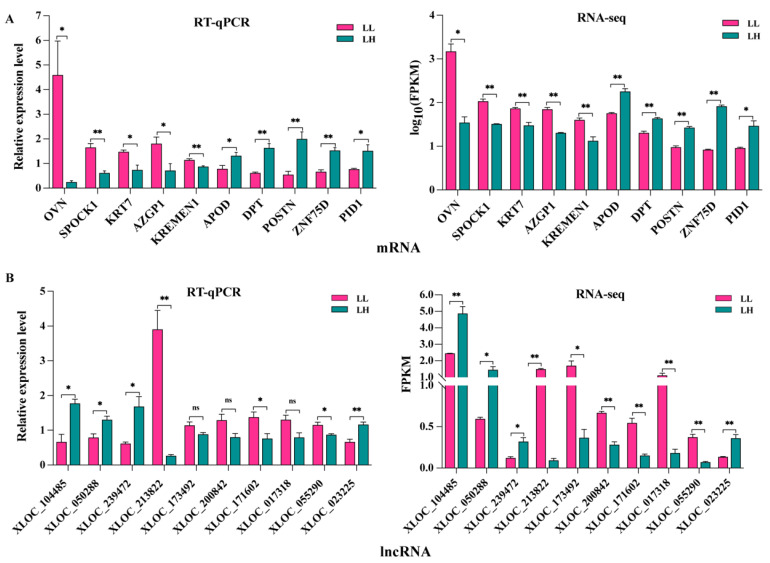
Validation of the RNA-seq data by RT-qPCR. (**A**) Selected mRNAs and (**B**) lncRNAs were validated by RT-qPCR and RNA-seq, respectively. ** *p* < 0.01, * *p* < 0.05, ns represent not significant. The RT-qPCR data are presented as relative gene expression. RNA-seq data are presented as fragments per kilobase of transcripts per million mapped reads (FPKM).

**Figure 4 animals-12-02823-f004:**
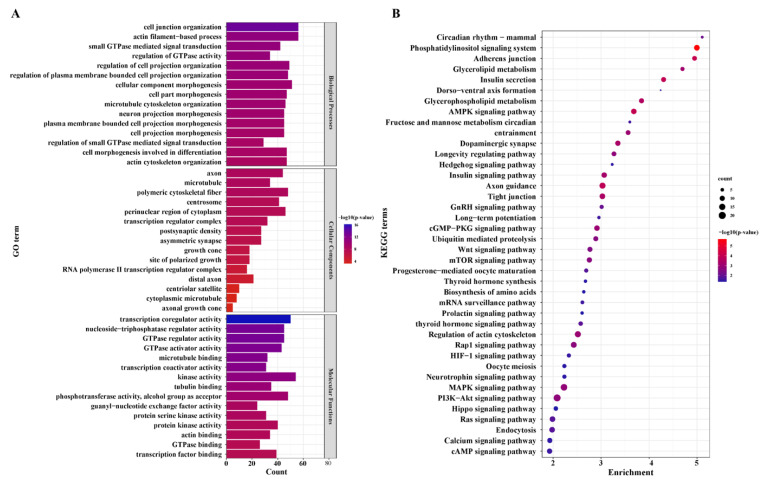
Go annotation and KEGG pathway enrichment of differentially expressed mRNAs (DEGs). (**A**) GO annotation of DEGs in LL vs. LH. (**B**) KEGG enrichment of DEGs in LL vs. LH.

**Figure 5 animals-12-02823-f005:**
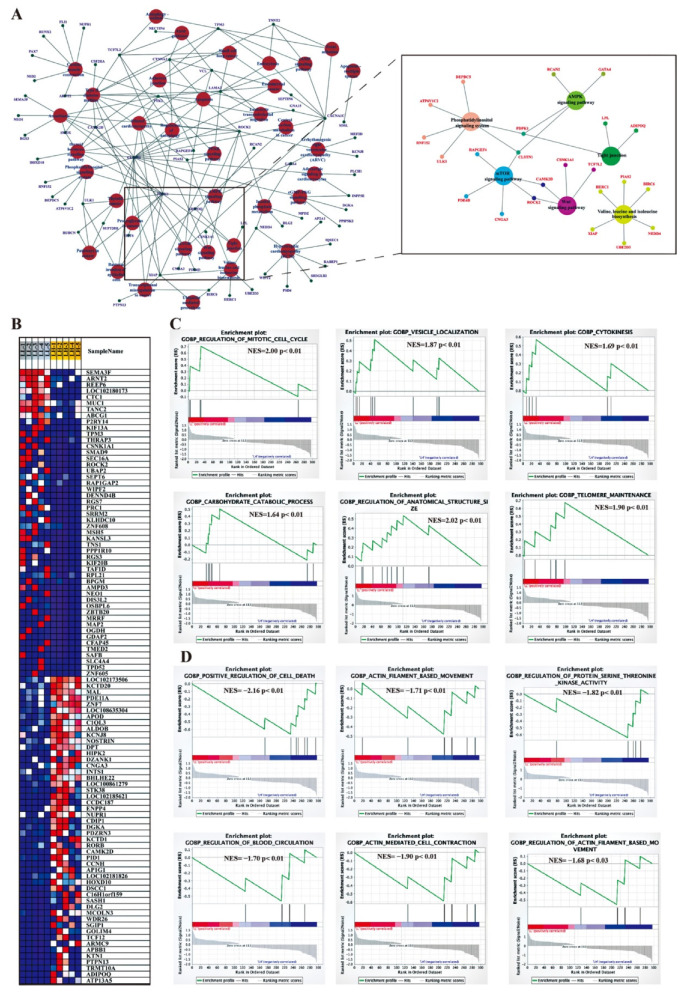
The function analysis of DELs-targeted DEGs. (**A**) Interaction and overlaps of associated genes among significantly enriched pathways. (**B**) The top 50 target genes feature for each phenotype group. GSEA was performed according to the expression of DELs-targeted DEGs in the LL groups (**C**) and LH group (**D**). NES: Normalized enrichment score. LL: Luteal phase with low-fecundity goats, LH: Luteal phase with high-fecundity goats.

**Figure 6 animals-12-02823-f006:**
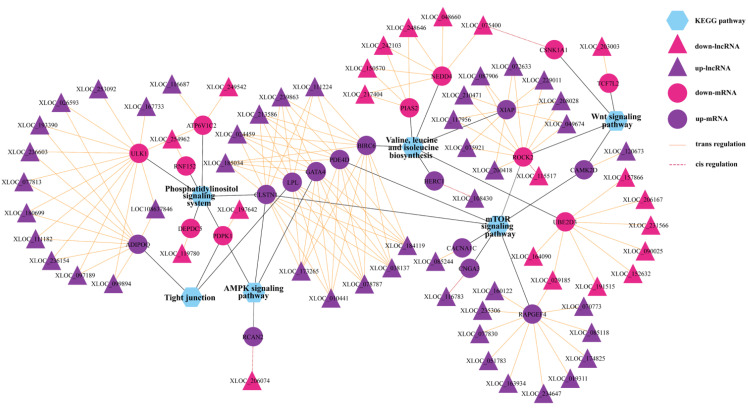
The interaction networks of DELs and their corresponding target genes and significant KEGG pathways. The solid and dotted lines are for trans- and cis-regulation functions. The hexagon, ellipses, and triangle represent the KEGG pathways, target mRNAs, and lncRNAs, respectively. The color pink and purple represent upregulated and downregulated, respectively.

**Table 1 animals-12-02823-t001:** The information of RNA-seq data.

Sample	Clean Reads	Clean Base (bp)	Length	Q20 (%)	Q30 (%)	GC (%)	Total Mapped (%)	Uniquely Mapped (%)
LH-1	53,564,145	16,069,243,500	150	98.10	94.45	46.70	97.11	93.06
LH-2	53,967,718	16,190,315,400	150	96.85	91.90	41.85	95.55	93.61
LH-3	53,642,740	16,092,822,000	150	98.55	95.40	43.10	97.20	94.62
LH-4	70,077,091	21,023,127,300	150	97.65	93.40	48.95	96.46	91.29
LH-5	53,146,812	15,944,043,600	150	98.10	94.35	46.05	97.12	93.33
LL-1	57,893,047	17,367,914,100	150	98.40	94.95	48.30	97.24	92.03
LL-2	64,129,075	19,238,722,500	150	98.40	95.10	46.70	97.30	93.89
LL-3	66,815,604	20,044,681,200	150	98.30	94.75	46.70	97.20	93.20
LL-4	53,717,853	16,115,355,900	150	98.20	94.60	47.10	97.23	92.67
LL-5	60,845,824	18,253,747,200	150	98.40	95.10	43.90	97.35	94.48

Note: LH represents high fecundity goats in the luteal phase; LL represents low fecundity goats in the luteal phase.

## Data Availability

All the data obtained from RNA-seq have been deposited in the Sequence Read Archive databases (SRA) of NCBI under accession numbers: SRX16002121-SRX16002130 (The accession number of BioProject is PRJNA854769). The URL is https://www.ncbi.nlm.nih.gov/sra/PRJNA854769.

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
