# Peer review of "Oviduct Transcriptomic Reveals the Regulation of mRNAs and lncRNAs Related to Goat Prolificacy in the Luteal Phase"

_animals, 2022, doi:10.3390/ani12202823_

Round 1
Reviewer 1 Report
In this paper, the authors compare the different expression of mRNA and lncRNA in the oviduct of the goat.
The study was well designed, and the results are clear and comprehensive. Overall, the paper turns out to be interesting and could add new knowledge in the field of reproduction.
One suggestion could be to include Fig.4 and Fig.7 in the supplemental materials.
Author Response
Response to Reviewer 1
In this paper, the authors compare the different expression of mRNA and lncRNA in the oviduct of the goat.
The study was well designed, and the results are clear and comprehensive. Overall, the paper turns out to be interesting and could add new knowledge in the field of reproduction.
Response: Thank you very much for your positive comments and suggestions.
One suggestion could be to include Fig.4 and Fig.7 in the supplemental materials.
Response: Thanks for your suggestions. We have added the Figure 4 into attachment as Figure S1. We think Figure 7 should appear in the text, because it is the only evidence that can show the relationship between key DELs and their corresponding target genes, after we have put the interaction diagrams of all DELs and target genes in the attachment. Please see the attachment.
Reviewer 2 Report
In this paper, RNA-seq were conducted on the oviducts of Yunshang black goats with high and low fertility, which provides a reference for improving the fertility and molecular of sheep. However, there are some problems that require refinements and revisions by the authors.
1. In the Introduction, some reports that have little relevance to the content of this study (e.g., lines 81~90) were advised to remove.
2. Table.1 should be considered as part of the supplementary materials, such as ASMT, ADAMTS1, and SSCP.
3. The full name of abbreviation should be listed when they were first mentioned, such as ASMT,ADAMTS1(line 69), and SSCP (line 70).
4. Italics should be considered to use when the abbreviations represented genes, such as DHX34 (line 33), LHX6 (line 33), and H19 (line 82).
5. Several references should be cited, such as line 76, line 92, line 100, and line 105.
6. In line 463, the authors mentioned ‘several lines of evidence indicate that lncRNAs are correlated with mammalian reproduction’. However, only one study on the correlation between lncRNA and women repeated implantation failure was cited here, which is clearly inconsistent with the description of the authors and more evidence should be provided.
7. Please check the spelling of the title ‘lncRAs’ and ‘mRAs’ in Supplementary Table S2 and S3. Additionally, authors should confirm whether these lncRNAs listed in Supplementary Table S2 are ‘differentially expressed lncRAs’ as the title describes.
Author Response
Response to Reviewer 2
In this paper, RNA-seq were conducted on the oviducts of Yunshang black goats with high and low fertility, which provides a reference for improving the fertility and molecular of sheep.
Response: Thank you very much for your positive and constructive comments and suggestions on the manuscript. We have tried our best to make a big improve about our manuscript according to your comments about the weaknesses in our manuscript, all details as follows.
However, there are some problems that require refinements and revisions by the authors.
1. In the Introduction, some reports that have little relevance to the content of this study (e.g., lines 81~90) were advised to remove.
Response: Thanks for your nice suggestion. We have removed some report that have little relevance to the content of this study.
- Gabory, A.; Jammes, H.; Dandolo, L. The H19 locus: role of an imprinted non-coding RNA in growth and development. Bi-oessays. 2010, 32, 473-480. doi: 10.1002/bies.200900170
- Ling, Y.H.; Xu, L.N.; Zhu, L.; Sui, M.H.; Zheng, Q.; Li, W.Y.; Liu, Y.; Fang, F.G.; Zhang, X.R. Identification and analysis of dif-ferentially expressed long non-coding RNAs between multiparous and uniparous goat (Capra hircus) ovaries. PLoS One. 2017, 12, e0183163. doi: 10.1371/journal.pone.0183163
- Nakagawa, S.; Shimada, M.; Yanaka, K.; Mito, M.; Arai, T.; Takahashi, E.; Fujita, Y.; Fujimori, T.; Standaert, L.; Marine, J.C.; et al. The lncRNA Neat1 is required for corpus luteum formation and the establishment of pregnancy in a subpopulation of mice. Development. 2014, 141, 4618-4627. doi: 10.1242/dev.110544
- Chen, S.; Guo, X.F.; He, X.Y.; Di, R.; Zhang, X.S.; Zhang, J.L.; Wang, X.Y.; Chu, M.X. Transcriptome analysis reveals differentially expressed genes and long non-coding RNAs associated with fecundity in sheep hypothalamus with different FecB genotypes. Front Cell Dev Biol. 2021, 9, 633747. doi: 10.3389/fcell.2021.633747
- Chen, S.; Guo, X.F.; He, X.Y.; Di, R.; Zhang, X.S.; Zhang, J.L.; Wang, X.Y.; Chu, M.X. Insight into pituitary lncRNA and mRNA at two estrous stages in Small Tail Han sheep with different FecB genotypes. Front Endocrinol. 2021, 12, 789564. doi: 10.3389/fendo.2021.789564
2. Table. 1 should be considered as part of the supplementary materials.
Response: Thanks for your comment. We have added the Table 1 into attachment as the Table S10.
3. The full name of abbreviation should be listed when they were first mentioned, such as ASMT,ADAMTS1(line 69), and SSCP (line 70).
Response: Thanks for your nice suggestion. All of those have been revised in the manuscript.
4. Italics should be considered to use when the abbreviations represented genes, such as DHX34 (line 33), LHX6 (line 33), and H19 (line 82).
Response: Thanks for your nice suggestion. All of those have been revised in the manuscript.
5. Several references should be cited, such as line 76, line 92, line 100, and line 105.
Response: Thanks for your nice suggestion. We have cited references in the new version manuscript. The details were shown in the revised manuscript.
References:
- Sun, Z.P.; Hong, Q.H.; Liu, Y.F.; He, X.Y.; Di, R.; Wang, X.Y.; Ren, C.H.; Zhang, Z.J.; Chu, M.X. Characterization of circular RNA profiles of oviduct reveal the potential mechanism in prolificacy trait of goat in the estrus cycle. Front. Physiol. 2022, 13, 990691. doi: 10.3389/fphys.2022.990691.
- Liu,L.; Fang, F. Long Noncoding RNA Mediated Regulation in Human Embryogenesis, Pluripotency, and Reproduction. Stem Cells Int. 2022, 2022, 8051717. doi: 10.1155/2022/8051717.
- Wang, J.J.; Niu, M.H.; Zhang, T.; Shen, W.; Cao, H.G. Genome-Wide Network of lncRNA–mRNA During Ovine Oocyte Development from Germinal Vesicle to Metaphase II in vitro. Front. Physiol. 2020, 11, 1019. doi.org/10.3389/fphys.2020.01019.).
- Wang, Y.; H, S.G.; Yao, G.X.; Zhu, Q.L., He, Y.Q., Lu, Y.; Qi, J.; Xu, R.; Ding, Y.; Li, J.X.; et al. A Novel Molecule in Human Cyclic Endometrium: LncRNA TUNAR Is Involved in Embryo Implantation. Front Physiol. 2020, 11, 587448. doi: 10.3389/fphys.2020.587448.
- Zhao, Z.F.; Zou, X.; Lu, T.T.; Deng, M.; Li, Y.K.; Guo, Y.Q.; Sun, B.L.; Liu, G.B.; Liu, D.W. Identification of mRNAs and lncRNAs Involved in the Regulation of Follicle Development in Goat. Front. Genet. 2020, 11, 589076. doi.org/10.3389/fgene.2020.589076.
- Li, S. and Winuthayanon, W. 2017. Oviduct: roles in fertilization and early embryo development. J. Endocrinol. 232: R1–R26. doi: 10.1530/JOE-16-0302.
6. In line 463, the authors mentioned ‘several lines of evidence indicate that lncRNAs are correlated with mammalian reproduction’. However, only one study on the correlation between lncRNA and women repeated implantation failure was cited here, which is clearly inconsistent with the description of the authors and more evidence should be provided.
Response: Thanks for your comment. We have cited three new references. The details were shown in the revised manuscript.
- Zhou, L.; Liu, J.S.; Wang, A.B.; Liu, Y.; Yu, H.; Ouyang, H.S.; Pang, D.X. Investigation of the lncRNA THOR in Mice Highlights the Importance of Noncoding RNAs in Mammalian Male Reproduction. Biomedicines. 2021, 9, 859. doi: 10.3390/biomedicines9080859.
- Wan, Z.; Yang, H.; Chen, P.Y.; Wang, Z.B.; Cai, Y.; Xiaolei Yao, X.L.; Wang, F.; Zhang, Y.L. The Novel Competing Endogenous Long Noncoding RNA SM2 Regulates Gonadotropin Secretion in the Hu Sheep Anterior Pituitary by Targeting the Oar-miR-16b/TGF-β/SMAD2 Signaling Pathway. Cells. 2022, 11, 985. doi: 10.3390/cells11060985.
- Hu, H.Y.; Fu, Y.F.; Zhou, B.; Li, Z.W.; Jia, Q. Long non-coding RNA TCONS_00814106 regulates porcine granulosa cell proliferation and apoptosis by sponging miR-1343. Mol Cell Endocrinol. 2021, 520, 111064. doi: 10.1016/j.mce.2020.111064.
7. Please check the spelling of the title ‘lncRAs’ and ‘mRAs’ in Supplementary Table S2 and S3. Additionally, authors should confirm whether these lncRNAs listed in Supplementary Table S2 are ‘differentially expressed lncRAs’ as the title describes.
Response: Thanks for your reminder. All of those have been revised in the manuscript and Supplementary Table S2 and S3.
Reviewer 3 Report
1.For acronyms that appear for the first time, the full name needs to be listed, and subsequent
writings only need to write abbreviations. The author also needs to revise the whole article carefully.
2. Some sentences in the text are not smooth, please check and modify.
3. In some places, more than or less than is more appropriate than > or <, please modify them.
4. in the manuscript, numbers, less than 10, should be replaced in English.
Abstract part: DEG and DEL should be replaced by DEGs and DELs, respectively. Please check in full article.
5. about the interpretation of CNCI, CPC2, and PLEK, Line 187-193, they were slightly cumbersome, it might be better to be deleted.
6. Statistical Analysis, only the statistical methods for RT-qPCR was listed ,but the sequencing data was not. Please add.
7. about the information of primers, it’s better to list their relevant parameters, such as TM.
8. about the summary of Sequencing Data, it’s better to descript the information in different groups.
9. in Table 2, for the LL-1, LL-2….., and LH-1, LH-2…., it’s better to add a note for them.
10. for Supplementary Table S2, its information is inconsistent with the description in manuscript.
11. for Figure 4, what’s the background database for the PPI? Which should be descripted in the M & M.
12. For Figure 6A, it was not present in the manuscript.
13. For Figure 7, what is the mean for different color?
14. a few reference was not cited appropriately, Which cannot support correctly the materials in the text.
15. In Figure 1. Numbers with 5 or more digits in the image should be separated by commas by thousands. e.g., “30000” should be “30,000”.
16. Line 209 and 211, “qPCR” instead of “RT-qPCR”.
17. In Figure 3. ***p<0.001 should be removed, caused none of the results in Figure 3 are p<0.001.
18. Line 312, “| log2 FC (fold change) | > 1” should subscript 2.
19. Line 315, “log2(FPKM)” should subscript 2.
20. Line 328, “Fig.3” instead of “Figure.3”.
Author Response
Response to Reviewer 3
Dear Reviewer/Prof.:
We are very grateful to you for the helpful comments and suggestions on our manuscript! We have revised the manuscript carefully according to your suggestions and replied each point that you raised as completely as possible. Please see the following itemized reply. Thanks a lot again!
1. For acronyms that appear for the first time, the full name needs to be listed, and subsequent writings only need to write abbreviations. The author also needs to revise the whole article carefully.
Response: Thanks for your nice suggestion. All of those have been revised in the manuscript.
2. Some sentences in the text are not smooth, please check and modify.
Response: Thanks for your comment. The manuscript has been revised by a native speaker.
3. In some places, more than or less than is more appropriate than > or <, please modify them.
Response: Thanks for your comment. We have revised those in the text.
4. in the manuscript, numbers, less than 10, should be replaced in English.
Abstract part: DEG and DEL should be replaced by DEGs and DELs, respectively. Please check in full article.
Response: Thank you for your reminder. All of those have been revised in the manuscript.
5. about the interpretation of CNCI, CPC2, and PLEK, Line 187-193, they were slightly cumbersome, it might be better to be deleted.
Response: Thanks for your comment. It has been revised in the manuscript.
6. Statistical Analysis, only the statistical methods for RT-qPCR was listed ,but the sequencing data was not. Please add.
Response: Thanks for your nice suggestion. The statistical method for the sequencing data has been listed in the “Statistical Analysis”.
7. about the information of primers, it’s better to list their relevant parameters, such as TM.
Response: Thanks for your comment. We have moved Table 1 to the attachment and added relevant information. Please see that in the attachment Table S10.
8. about the summary of Sequencing Data, it’s better to descript the information in different groups.
Response: Thanks for your nice suggestion. It has been revised in the manuscript.
9. in Table 2, for the LL-1, LL-2….., and LH-1, LH-2…., it’s better to add a note for them.
Response: Thank you for your reminder. We have added the note under the Table 1 in the text.
10. for Supplementary Table S2, its information is inconsistent with the description in manuscript.
Response: Thank you for your reminder. It has been revised in the Supplementary Table S2. Please see that in the attachment.
11. for Figure 4, what’s the background database for the PPI? Which should be descripted in the M & M.
Response: Thanks for your nice suggestion. The background database of STRING was Ovis aries, we have described in the M & M.
12. For Figure 6A, it was not present in the manuscript.
Response: Thanks for your comment. We have changed Fugure 6A to Figure 5A in the new version, and is presented in the manuscript.
13. For Figure 7, what is the mean for different color?
Response: Thanks for your comment. We have changed Fugure 7 to Figure 6 in the new version, and is presented in the manuscript. For Figure 7, the color pink and purple denote upregulated and downregulated, respectively, which have been supplemented under the Figure 6.
14. a few reference was not cited appropriately, Which cannot support correctly the materials in the text.
Response: Thanks for your comment. We have removed, changed, and added some reference that have little relevance to the content of this study.
References that were removed in the old version:
- Gabory, A.; Jammes, H.; Dandolo, L. The H19 locus: role of an imprinted non-coding RNA in growth and development. Bi-oessays. 2010, 32, 473-480. doi: 10.1002/bies.200900170.
- Ling, Y.H.; Xu, L.N.; Zhu, L.; Sui, M.H.; Zheng, Q.; Li, W.Y.; Liu, Y.; Fang, F.G.; Zhang, X.R. Identification and analysis of dif-ferentially expressed long non-coding RNAs between multiparous and uniparous goat (Capra hircus) ovaries. PLoS One. 2017, 12, e0183163. doi: 10.1371/journal.pone.0183163.
- Nakagawa, S.; Shimada, M.; Yanaka, K.; Mito, M.; Arai, T.; Takahashi, E.; Fujita, Y.; Fujimori, T.; Standaert, L.; Marine, J.C.; et al. The lncRNA Neat1 is required for corpus luteum formation and the establishment of pregnancy in a subpopulation of mice. Development. 2014, 141, 4618-4627. doi: 10.1242/dev.110544.
- Chen, S.; Guo, X.F.; He, X.Y.; Di, R.; Zhang, X.S.; Zhang, J.L.; Wang, X.Y.; Chu, M.X. Transcriptome analysis reveals differentially expressed genes and long non-coding RNAs associated with fecundity in sheep hypothalamus with different FecB genotypes. Front Cell Dev Biol. 2021, 9, 633747. doi: 10.3389/fcell.2021.633747.
- Chen, S.; Guo, X.F.; He, X.Y.; Di, R.; Zhang, X.S.; Zhang, J.L.; Wang, X.Y.; Chu, M.X. Insight into pituitary lncRNA and mRNA at two estrous stages in Small Tail Han sheep with different FecB genotypes. Front Endocrinol. 2021, 12, 789564. doi: 10.3389/fendo.2021.789564.
- Chen, M.Y.; Liao, G.D.; Zhou, B.; Kang, L.N.; He, Y.M.; Li, S.W. Genome-wide profiling of long noncoding RNA expression 686 patterns in women with repeated implantation failure by RNA sequencing. Reprod Sci. 2019, 26, 18-25. doi: 687 10.1177/1933719118756752.
References that were changed or added in the new version:
- Wang, J.J.; Niu, M.H.; Zhang, T.; Shen, W.; Cao, H.G. Genome-Wide Network of lncRNA–mRNA During Ovine Oocyte Development from Germinal Vesicle to Metaphase II in vitro. Front. Physiol. 2020, 11, 1019. doi.org/10.3389/fphys.2020.01019.
- Wang, Y.; Guo, Y.X.; Duan, C.H.; Yang, R.C.; Zhang, L.C.; Liu, Y.Q.; Zhang, Y.J. Long Non-Coding RNA GDAR Regulates Ovine Granulosa Cells Apoptosis by Affecting the Expression of Apoptosis-Related Genes. Int J Mol Sci. 2022, 23(9), 5183. doi: 10.3390/ijms23095183.
- Wang, Y.; H, S.G.; Yao, G.X.; Zhu, Q.L., He, Y.Q., Lu, Y.; Qi, J.; Xu, R.; Ding, Y.; Li, J.X.; et al. A Novel Molecule in Human Cyclic Endometrium: LncRNA TUNAR Is Involved in Embryo Implantation. Front Physiol. 2020, 11, 587448. doi: 10.3389/fphys.2020.587448.)
- Zhao, Z.F.; Zou, X.; Lu, T.T.; Deng, M.; Li, Y.K.; Guo, Y.Q.; Sun, B.L.; Liu, G.B.; Liu, D.W. Identification of mRNAs and lncRNAs Involved in the Regulation of Follicle Development in Goat. Front. Genet. 2020, 11, 589076. doi.org/10.3389/fgene.2020.589076.
- Zhou, L.; Liu, J.S.; Wang, A.B.; Liu, Y.; Yu, H.; Ouyang, H.S.; Pang, D.X. Investigation of the lncRNA THOR in Mice Highlights the Importance of Noncoding RNAs in Mammalian Male Reproduction. Biomedicines. 2021, 9, 859. doi: 10.3390/biomedicines9080859.
- Wan, Z.; Yang, H.; Chen, P.Y.; Wang, Z.B.; Cai, Y.; Xiaolei Yao, X.L.; Wang, F.; Zhang, Y.L. The Novel Competing Endogenous Long Noncoding RNA SM2 Regulates Gonadotropin Secretion in the Hu Sheep Anterior Pituitary by Targeting the Oar-miR-16b/TGF-β/SMAD2 Signaling Pathway. Cells. 2022, 11, 985. doi: 10.3390/cells11060985.
- Hu, H.Y.; Fu, Y.F.; Zhou, B.; Li, Z.W.; Jia, Q. Long non-coding RNA TCONS_00814106 regulates porcine granulosa cell proliferation and apoptosis by sponging miR-1343. Mol Cell Endocrinol. 2021, 520, 111064. doi: 10.1016/j.mce.2020.111064.
15. In Figure 1. Numbers with 5 or more digits in the image should be separated by commas by thousands. e.g., “30000” should be “30,000”.
Response: Thanks for your nice suggestion. We have changed it in the new version.
16. Line 209 and 211, “qPCR” instead of “RT-qPCR”.
Response: Thanks for your nice suggestion. We have changed it in the new version.
17. In Figure 3. ***p<0.001 should be removed, caused none of the results in Figure 3 are p<0.001.
Response: Thanks for your nice suggestion. It has been removed in the manuscript.
18. Line 312, “| log2 FC (fold change) | > 1” should subscript 2.
Response: Thanks for your nice suggestion. It has been revised in the manuscript.
19. Line 315, “log2(FPKM)” should subscript 2.
Response: Thanks for your nice suggestion. It has been revised in the manuscript.
20. Line 328, “Fig.3” instead of “Figure.3”.
Response: Thanks for your nice suggestion. It has been revised in the manuscript.
Reviewer 4 Report
Evaluation
An attempt was made to identify the expression of mRNAs and lncRNAs related to prolificacy in oviduct of goat.
The RNA-seq studies were conducted on oviduct at the luteal phase from high- and low-fecundity goats. 2,023 differentially expressed mRNA and 377 differentially expressed lncRNA transcripts were screened, and 2,109 regulated lncRNA‒mRNA pairs were identified.
Several candidate mRNAs and lncRNAs with potential functions were identified by functional bioinformatics analyses.
General comments:
In my opinion, the introduction has to be implemented also with some recent lacking references, better framing the topic and the background, clarifying the aim and the applicability of the work.
The methodology and statistical analysis are very robust and experimental design is correct.
In the discussion, the authors relate their results sufficiently to the results of other authors. it would be appropriate to present in more detail the fundamental and applied significance of your results.
Author Response
Response to Reviewer 4
Evaluation
An attempt was made to identify the expression of mRNAs and lncRNAs related to prolificacy in oviduct of goat.
The RNA-seq studies were conducted on oviduct at the luteal phase from high- and low-fecundity goats. 2,023 differentially expressed mRNA and 377 differentially expressed lncRNA transcripts were screened, and 2,109 regulated lncRNA‒mRNA pairs were identified.
Several candidate mRNAs and lncRNAs with potential functions were identified by functional bioinformatics analyses.
Response: Thanks so much for your comments concerning our manuscript, which are all valuable and very helpful for revising and improving our paper! Following those comments and suggestions, we have made correction carefully in the manuscript. Now, those revised portion were marked up using the “Track Changes” in this word file. Thanks again for your help and suggestions!
General comments:
1. In my opinion, the introduction has to be implemented also with some recent lacking references, better framing the topic and the background, clarifying the aim and the applicability of the work.
Response: Thanks for your comment. We have cited some recent references in the new version.
References:
- Sun, Z.P.; Hong, Q.H.; Liu, Y.F.; He, X.Y.; Di, R.; Wang, X.Y.; Ren, C.H.; Zhang, Z.J.; Chu, M.X. Characterization of circular RNA profiles of oviduct reveal the potential mechanism in prolificacy trait of goat in the estrus cycle. Front. Physiol. 2022, 13, 990691. doi: 10.3389/fphys.2022.990691
- Liu,L.; Fang, F. Long Noncoding RNA Mediated Regulation in Human Embryogenesis, Pluripotency, and Reproduction. Stem Cells Int. 2022, 2022, 8051717. doi: 10.1155/2022/8051717.
- Wang, J.J.; Niu, M.H.; Zhang, T.; Shen, W.; Cao, H.G. Genome-Wide Network of lncRNA–mRNA During Ovine Oocyte Development from Germinal Vesicle to Metaphase II in vitro. Front. Physiol. 2020, 11, 1019. doi.org/10.3389/fphys.2020.01019.
- Wang, Y.; Guo, Y.X.; Duan, C.H.; Yang, R.C.; Zhang, L.C.; Liu, Y.Q.; Zhang, Y.J. Long Non-Coding RNA GDAR Regulates Ovine Granulosa Cells Apoptosis by Affecting the Expression of Apoptosis-Related Genes. Int J Mol Sci. 2022, 23(9), 5183. doi: 10.3390/ijms23095183.
- Wang, Y.; H, S.G.; Yao, G.X.; Zhu, Q.L., He, Y.Q., Lu, Y.; Qi, J.; Xu, R.; Ding, Y.; Li, J.X.; et al. A Novel Molecule in Human Cyclic Endometrium: LncRNA TUNAR Is Involved in Embryo Implantation. Front Physiol. 2020, 11, 587448. doi: 10.3389/fphys.2020.587448
- Zhao, Z.F.; Zou, X.; Lu, T.T.; Deng, M.; Li, Y.K.; Guo, Y.Q.; Sun, B.L.; Liu, G.B.; Liu, D.W. Identification of mRNAs and lncRNAs Involved in the Regulation of Follicle Development in Goat. Front. Genet. 2020, 11, 589076. doi.org/10.3389/fgene.2020.589076.
2. The methodology and statistical analysis are very robust and experimental design is correct.
Response: We thank you very much for your positive comments on our manuscript.
3. In the discussion, the authors relate their results sufficiently to the results of other authors. it would be appropriate to present in more detail the fundamental and applied significance of your results.
Response: Thanks for your comment. The fundamental and applied significance has been supplemented in the conclusion.
Reviewer 5 Report
The manuscript titled Transcriptome differential expression profiling reveals the mRNA and lncRNAs that related to prolificacy in oviduct of goat at the luteal phase collected oviductal tissue from 5 low and 5 high prolificacy estrus synchronized goats. Although the title is cumbersome and should be rethought, the manuscript is mostly well written. This data should be interpreted with caution since it is merely associative. More detail on the females should be given (ie parity, age, how prolificacy was established). There is nothing to confirm the goats were in the luteal phase, and this is concerning since many of the genes are regulated by steroids. Although significant differences are reported in Figure 3, the relative differences are small. Pathways mapped (Figure 7) to the differentially expressed genes are ubiquitous so it is not surprising that these pathways are potentially influenced.
Author Response
Response to Reviewer 5
The manuscript titled Transcriptome differential expression profiling reveals the mRNA and lncRNAs that related to prolificacy in oviduct of goat at the luteal phase collected oviductal tissue from 5 low and 5 high prolificacy estrus synchronized goats. Although the title is cumbersome and should be rethought, the manuscript is mostly well written. This data should be interpreted with caution since it is merely associative. More detail on the females should be given (ie parity, age, how prolificacy was established). There is nothing to confirm the goats were in the luteal phase, and this is concerning since many of the genes are regulated by steroids. Although significant differences are reported in Figure 3, the relative differences are small. Pathways mapped (Figure 7) to the differentially expressed genes are ubiquitous so it is not surprising that these pathways are potentially influenced.
Response: Thank you very much for the reviewer’s friendly suggestion.
1. Although the title is cumbersome and should be rethought, the manuscript is mostly well written.
Response: Thank you very much for the helpful suggestion! The title has been modified following your suggestion. “Oviduct Transcriptomic Reveals the Regulation of mRNAs and lncRNAs Related to Prolificacy of Goat at the Luteal Phase”
2. More detail on the females should be given (ie parity, age, how prolificacy was established).
Response: Thanks for your comment. We have added these information in the manuscript. Ten female goats (3 years old) were selected and grouped into two groups, including high fecundity goats in the luteal phase (LH) (n = 5, average kidding number 3.4±0.42) and low fecundity goats (n = 5, average kidding number 1.8±0.27, LL groups) based on their kidding number records. In addition, we further established the prolificacy of the goats selected for this study, based on the number of ovulations and the number of corpora luteum (the average corpus luteum number of LH and LL groups was 3.6 ± 0.40 and 1.6 ± 0.24, respectively).
3. There is nothing to confirm the goats were in the luteal phase, and this is concerning since many of the genes are regulated by steroids.
Response: Thank you very much for evaluating our work. It has been revised in “Materials and Methods”. We have counted the average number of corpus luteum produced by high- and low-fecundity goats as 3.6±0.40 and 1.6 ± 0.24, respectively, which is an important evidence for us to to confirm the goats were in the luteal phase.
4. Although significant differences are reported in Figure 3, the relative differences are small.
Response: Thanks for your comment. In this study, 10 differentially expressed mRNA and 10 differentially expressed lncRNA transcripts were selected for RT-qPCR to verify the accuracy of RNA-seq results. Our results show that the expression of the selected transcripts is consistent with that of RNA-seq. Although there are some transcripts with relatively small differences, it can still show the reliability of the results, because their expression trends are consistent.
5. Pathways mapped (Figure 7) to the differentially expressed genes are ubiquitous so it is not surprising that these pathways are potentially influenced.
Response: Thanks for your comment. As you said, there are many pathways mapped to differentially expressed genes, but several key pathways related to reproduction and oviduct function were selected for discussion, with the aim of screening some key genes and mRNA-lncRNA interactions relationship associated with our study.
Round 2
Reviewer 2 Report
The authors have replied to the reviewer's concerns.
Reviewer 5 Report
There remains some minor editing.